# Spike-timing dependent plasticity partially compensates for neural delays in a multi-layered network of motion-sensitive neurons

**Charlie M. Sexton**[1]*, **Anthony N. Burkitt**[2,3], **Hinze Hogendoorn**[1,4]

**1** Melbourne School of Psychological Sciences, The University of Melbourne, Victoria, Australia, **2** Department of Biomedical Engineering, The University of Melbourne, Victoria, Australia, **3** Graeme Clark Institute for Biomedical Engineering, The University of Melbourne, Victoria, Australia, **4** School of Psychology and Counselling, Queensland University of Technology, Queensland, Australia

* charlie.sexton@unimelb.edu.au

**Data Availability Statement:** Data and code used for network training and analyses are available at

## Abstract

The ability of the brain to represent the external world in real-time is impacted by the fact that neural processing takes time. Because neural delays accumulate as information progresses through the visual system, representations encoded at each hierarchical level are based upon input that is progressively outdated with respect to the external world. This 'representational lag' is particularly relevant to the task of localizing a moving object–because the object's location changes with time, neural representations of its location potentially lag behind its true location. Converging evidence suggests that the brain has evolved mechanisms that allow it to compensate for its inherent delays by extrapolating the position of moving objects along their trajectory. We have previously shown how spike-timing dependent plasticity (STDP) can achieve motion extrapolation in a two-layer, feedforward network of velocity-tuned neurons, by shifting the receptive fields of second layer neurons in the opposite direction to a moving stimulus. The current study extends this work by implementing two important changes to the network to bring it more into line with biology: we expanded the network to multiple layers to reflect the depth of the visual hierarchy, and we implemented more realistic synaptic time-courses. We investigate the accumulation of STDP-driven receptive field shifts across several layers, observing a velocity-dependent reduction in representational lag. These results highlight the role of STDP, operating purely along the feedforward pathway, as a developmental strategy for delay compensation.

## Author summary

To promote survival, organisms must be able to quickly react to things in the world that pose a threat. For example, the appearance of a swooping bird in your peripheral vision may cause you to quickly duck your head. But the brain activity underlying our perception of the threat does not happen instantaneously–it takes time for sensory information to be processed–meaning that our perceptual experience has an inherent lag. Yet, we clearly possess a remarkable ability to interact with fast moving objects, such as when catching a

https://osf.io/fczky/ (DOI: 10.17605/OSF.IO/FCZKY).

**Funding:** This work was supported by an Australian Research Council (ARC) Discovery Project Grant [DP180102268] and Future Fellowship Grant [FT200100246] to H.H., and an Australian Government Research Training Program (RTP) Scholarship to C.M.S. ANB & HH acknowledge funding support by the Australian Government through the Australian Research Council's Discovery Projects funding scheme [DP220101166]. The funders had no role in study design, data collection and analysis, decision to publish, or preparation of the manuscript.

**Competing interests:** The authors have declared that no competing interests exist.

cricket ball. Research has looked at how the brain may compensate for internal delays by shifting its representations of moving objects forward. Here, we look at how this can be achieved by plasticity at connections between different areas of the visual system. Our results show that a certain type of plasticity known to occur throughout the brain, called spike-timing dependent plasticity, can cause the represented location of moving objects to be shifted forward at each area. These forward-shifts accumulate at each level of our model network, meaning that the highest layers (which are subject to long processing delays) become capable of accurately representing the moving object closer to its realtime position.

## Introduction

The ability of the brain to represent the external world in real-time is constrained by the fact that neural processing takes time. In the case of vision, the majority of afferent information is first encoded by the retina and sent to the lateral geniculate nucleus (LGN) of the thalamus, before progressing to the primary visual cortex (V1) and on to the various levels of the cortical processing hierarchy. The time it takes for this information transfer to occur is limited by the speed of axonal and dendritic conduction [1,2]. Further delays accumulate due to the fact that individual neurons reach firing threshold only after receiving a sequence of inputs, as a result of the integration time of the membrane potential [3]. For example, onset latencies of LGN neurons to a flashed visual stimulus are on the order of 30 to 70ms, and can be as high as ~90ms in V1 [4]. Because these delays accumulate as information progresses through the visual system, representations encoded at each hierarchical level are based upon input that is progressively outdated with respect to the external world. This is particularly relevant to the task of localizing a moving object–because the object's location changes with time, neural representations of its location potentially lag behind its true location.

It has been suggested that mechanisms within the visual pathway serve to compensate for these neural delays, by extrapolating the position of moving objects along the path of motion [5,6], and thereby reducing or eliminating the degree of lag between the object's veridical position and its internal representation–a notion we will hereafter refer to as 'representational lag'. In support of this idea, neurophysiological studies have shown that neural representations of moving stimuli are shifted forward along the motion trajectory, at the level of the retina [7,8], V1 [9–12] and V4 [13]. Additionally, recent human neuroimaging data have shown that representations of moving objects align with the realtime position of a moving stimulus, despite neural delays [14]. In addressing the precise mechanisms underlying these position shifts, many studies have focused on the role of local, horizontal connections among retinotopic regions [9,10,12]. It has been shown, however, that asymmetries among feedforward connections between motion-sensitive neurons can also produce shifts in population activity in the forward direction of motion [11,15].

Fu et al [11]. observed that the receptive fields of neurons in cat V1 shifted in the direction opposite to a motion signal, and showed how such a shift can be produced by spike-timing dependent plasticity (STDP; [16,17]). STDP is a form of long-term plasticity in which a synapse is strengthened or weakened based on the relative timing of pre- and post-synaptic potentials; pre-synaptic spikes occurring within a short time frame before a post-synaptic spike will lead to long-term potentiation, while those occurring shortly after a post-synaptic spike lead to long-term depression. To illustrate how STDP can shift the receptive field of motion-sensitive neurons, consider a simple case in which a target neuron at one hierarchical level receives

feedforward input from several cells at a lower level that each have neighbouring receptive field centers. During motion, cells at the lower level will be successively excited, driving activity in the target cell. In the case of left-to-right motion, cells on the left side will tend to fire before the target neuron, whereas those on the right side will tend to fire after. STDP will therefore tend to strengthen the connections of cells on the left and weaken connections of cells on the right, leading to a shift of the target neurons receptive field in the direction opposite to motion [11,15]. As a result, that neuron will begin firing earlier in subsequent motion traversals. This, in turn, shifts the position encoded by a population of neurons in the forward direction of motion.

Burkitt et al [15]. showed how such a receptive-field shift could serve to compensate for neural delays, at least in part, by simulating a two-layer, feedforward network encompassing several velocity-tuned subpopulations. They presented moving stimuli to the network across a range of velocities, and allowed the connection weights to evolve according to the STDP learning rule. They showed that the degree of receptive-field shift among second-layer neurons scaled with the velocity of the stimulus, i.e. the degree of forward shift in the encoded position of the object increased with the degree of representational lag that the network had to contend with. While these results show how STDP can produce extrapolatory positional representations akin to those seen in biology, their model had two important limitations.

Firstly, the visual processing system comprises many hierarchical levels beyond the two simulated in Burkitt et al [15]. Each additional level increases the overall potential for representational lag to be incurred in the system. Other modelling work has implemented STDP as a learning mechanism in multilayer spiking neural networks (SNNs), with particular focus on its capacity to generate motion direction [18] and velocity [19] selectivity in simulated neurons. Although Burkitt et al [15]. did not address how velocity-sensitivity was formed among the neurons in their simulation, they showed how STDP can alter the structure of feedforward connectivity *between* those neurons, in such a way that downstream neurons begin to respond earlier to motion stimuli. To our knowledge, no work to date has addressed the combined effect of STDP-driven receptive field shifts [11,15] across multiple levels of a hierarchical network. Secondly, the network simulated in Burkitt et al [15]. implemented a simplified, biologically-implausible instantaneous synaptic transmission model. Using a more realistic model of the synaptic membrane potential would potentially introduce additional delays (due to temporal integration), and it is not known how this integration process will influence the effect of STDP in the network. It is therefore of interest to determine the extent to which STDP can facilitate reduction of representational lag to the extent that it is encountered in a multilayer network with transmission and integration delays.

The current project extends the modelled network in Burkitt et al [15]. to address these shortcomings. To do so, we have expanded the model to six layers, to better represent the depth of the visual processing hierarchy. In addition, we introduce a membrane time constant to the post-synaptic potential, bringing the synaptic behaviour more in line with biology. We show that when these changes are implemented, the effects of STDP accumulate systematically throughout the network, reducing the overall lag between the realtime position of a moving stimulus and its neural representation.

## Methods

### Neural model

We simulated a network of spiking neurons *in silico* in the same way as Burkitt et al [15]., with the following exceptions. In the modified network, each layer consists of $N_n$ neurons, spatially tuned to equally distributed points across a circular interval [0, 1]. There are $N_l$ layers in total,

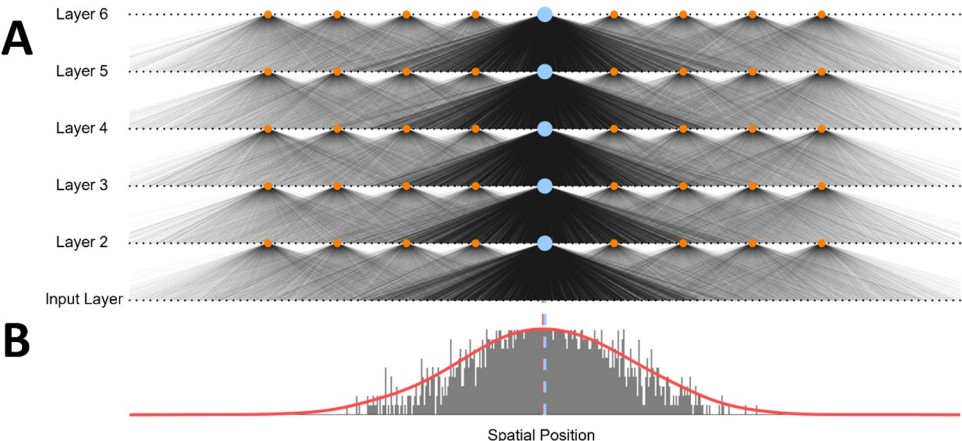

**Fig 1. Network connections in an untrained network. A.** Subset of neural connections prior to STDP learning procedure. Higher-layer connections (layers 2 to 6) for a single neural index (blue circles) are shown with increased opacity to highlight a typical set of connections. Neighbouring neurons (orange circles) have overlapping receptive fields. Only the connections for every 50th neuron within the region plotted are shown to maintain clarity. Relative opacity levels indicate starting weights. **B.** Distribution of feedforward weights for central neuron at layer two. The peak of the probability density function (dotted red line) is in line with the location of the neuron, prior to any STDP.

and each neuron has a limited number of feed-forward, excitatory connections to neurons in the layer above (Fig 1). The probability of such a connection follows a Gaussian distribution with width $\sigma_w$. The weights of connections are described by a $N_l$ x $N_n$ x $N_n$ matrix $W$ which encompasses all connections between neurons for each higher layer. The starting weight of each connection is initialized at $w_{init} = \frac{1}{N_w}$ where $N_w$ is the average number of connections per layer. A small randomness is then applied (uniformly distributed in the range +/- 0.025). Each layer has equal spatial separation and an equal, fixed transmission delay time $t_{delay}$. The network thus described is replicated across $N_v$ velocity-tuned subpopulations, with activity in each subpopulation determined in an all-or-none fashion depending on the input stimulus; only the subpopulation tuned to a particular stimulus velocity will be activated, all others remain inactive. We also include a zero-velocity/static subpopulation of neurons that are activated by stationary input. This 'static network' can be regarded as a subpopulation of neurons tuned to stimulus position but with no velocity tuning. In this arrangement, the network as a whole is comprised of $N_{v+1}$ subnetworks with their own separate connection weights. Note that we do not address the formation of motion-sensitivity in this network, but rather treat each subpopulation as having already acquired tuning for a particular input velocity, prior to the STDP learning procedure described here.

**Input layer.** We consider here a simple point-like stimulus, which is encoded by the spiking rate of neurons in the first (input) layer, $r_j^1(t)$, following a Gaussian distribution centered on the stimulus location with width $\sigma_p$. Neurons in the input layer have a baseline firing rate $r_b$, and the magnitude of the stimulus is set to $20r_b$. Spikes are generated stochastically according to a Poisson process [20], in which the probability of a neuron $i$ firing in the time-interval $\Delta t$ is $r_i \Delta t$. Spiking activity generated in the input layer reaches the second layer after a transmission delay time $t_{delay}$.

**Higher layers.** Each higher layer $l$ receives input only from the layer below it. Firing rates for a neuron $j$ ($j = 1, \ldots, N_n$) in this layer at a given timepoint $t$ are determined by

$$r_j^l(t) = r_j^l(t - \Delta t)e^{-\frac{\Delta t}{\tau_m}} + \frac{\Delta t}{\tau_m}I_j^l(t), \tag{1}$$

where $\Delta t$ is the length of the discrete timestep used in the simulation (1 ms), $\tau_m$ the passive membrane time constant that determines the time scale of integration of the synaptic input (i.e., the post-synaptic potential), and $I_j^l(t)$ is the input to this neuron, given by

$$I_j^l(t) = \frac{1}{\Delta t} \sum_i W_{ji}^{l-1} S_i^{l-1}(t - t_{\text{delay}}),$$ (2)

where $W_{ji}^{l-1}$ are the feedforward weights from the layer below and $S_i^{l-1}$ are the (delayed) spikes from the layer below.

## Network learning

At each training timepoint, weights are updated according to a balanced STDP function with equal time constants for both potentiation and depression. A growth ceiling is imposed such that weights within a given layer cannot grow beyond more than 3 times the maximum starting weight for that layer. Weights cannot go below zero. The change in weights at each timepoint is described by

$$\Delta w_{ji}^l = p_t F_{\text{STDP}}(t_j^l - t_i^l - t_{\text{delay}})$$ (3)

where $p_t$ is the learning rate for that timestep (note that a variable learning rate was used: see *Numerical Simulation*), and $t_j^l$ and $t_i^l$ are the spike times for the post-synaptic and pre-synaptic cells respectively. A change in weight will only occur when the time difference $\Delta t$ between pre- and post-synaptic spikes is inside the STDP time-window:

$$F_{\text{STDP}}(\Delta t) = \begin{cases} c_p e^{-\frac{\Delta t}{\tau_p}}, \Delta t > 0 \\ 0, \Delta t = 0 \\ -c_d e^{\frac{\Delta t}{\tau_d}}, \Delta t < 0, \end{cases}$$ (4)

where $c_p$ and $c_d$ are the coefficients of potentiation and depression, and $\tau_p$ and $\tau_d$ are the time constants for potentiation and depression respectively. Table 1 lists the parameter values used in the numerical simulation.

**Table 1. Network Parameter Values Used in the Numerical Simulation.**

| Name | Value | Description |
|---|---|---|
| $N_n$ | 2000 | Number of neurons per layer |
| $N_l$ | 6 | Number of layers |
| $N_v$ | 10 | Number of velocity-tuned subpopulations |
| $\sigma_w$ | 1/32 | Parameter determining width of anatomical connections |
| $\sigma_p$ | 1/32 | Parameter determining width of input layer activations |
| $r_b$ | 5 Hz | Baseline firing rate at input layer |
| $t_{\text{delay}}$ | 20 ms | Neural transmission delay between layers |
| $c_p$ | 1 | Coefficient of STDP potentiation |
| $c_d$ | 1 | Coefficient of STDP depression |
| $\tau_p$ | 20 ms | STDP potentiation time constant |
| $\tau_d$ | 20 ms | STDP depression time constant |
| $\Delta t$ | 1 ms | Length of discrete timesteps |
| $\tau_m$ | 10 ms | Membrane time constant |

**Normalization.** Weights are normalized at random timepoints during the training procedure, in order to maintain the sum of weights for each layer at its starting value of ~1. Normalization occurs with a probability of 0.01 at each timestep (occurring on average once in every 100 timesteps), according to:

$$W_{ji}^l = \frac{W_{ji.}^l}{\sum_i W_{ji}^l} \qquad (5)$$

## Numerical simulation

**Training procedure.** The training procedure takes place over $N_t$ = 20,000 timesteps (equivalent to 20 simulated seconds). Although this duration is shorter than a true biological training window is likely to be, our analysis focuses on the asymptotic properties of the network, and the results should therefore be independent of how long the network takes to reach that asymptote. Choosing a relatively high learning rate and short training time allowed us to save computation time. Following initialization of the network weights, a moving input stimulus traverses the input for the full 20 seconds, evoking activity in the subpopulation tuned to the stimulus velocity. Weights within that population evolve according to STDP at each timestep–no STDP is applied to the inactive subpopulations. A variable STDP learning-rate $p(t)$ is used that ramps linearly from 0 to 0.0004 across the first $N_t/4$ timesteps, and then remains fixed at this value for the rest of the training. Training is conducted independently across different levels of velocity and different values of the membrane time constant. For each combination, 10 independent training runs are conducted with randomly determined starting points of motion, and the weights of each completed run are averaged to produce the final set of weights for that network. The range of velocities used in the simulation is 0.1 to 1 cycles per second in increments of 0.1. In addition, we include a set of higher velocities at greater intervals (2, 3, 4 and 5 cycles/s), in order to compare results with the range of velocities tested in Burkitt et al [15]. while reducing computational load.

As mentioned, neurons in the static network are regarded as being responsive to localised, stationary inputs, rather than being activated by a specific velocity. Although these neurons would be subject to STDP in a biological setting, a balanced STDP function and non-directional inputs would result in spatially symmetric connection weights, essentially replicating the initial state of the network. We therefore do not perform any training procedure for this network–the weights for this subpopulation are initialised as described above and do not undergo any STDP learning. The membrane time constant $\tau_m$ is set at 10 ms for the main analyses.

**Spike-collection stage.** After the process of STDP learning, spiking responses are compared between the (trained) velocity-tuned networks and the (untrained) static network. Specifically, responses of each velocity-tuned network to motion input at their preferred velocity are compared with responses of the static network to stationary flashes. The spike collection procedure proceeds as follows: for each velocity, the final set of trained weights for the corresponding subpopulation are loaded. 200 trials are performed, during which a motion stimulus starts in a random location and moves at the preferred velocity for the duration of the trial (700 ms; Fig 2B). The stimulus has the same parameters as those used in training. At each timestep, activity propagates throughout the network according to the same model used in training, and a probability density function (PDF) is fitted to the distribution of spiking activity for each layer (Fig 2C, red lines; see *Finding Peak of Spiking Response*). The peak of the PDF at each layer is taken as the positional representation of stimulus location at that layer and timepoint. No further STDP or normalization is performed during this simulation. At the halfway point of each trial, a stationary stimulus is presented at the same location as the motion

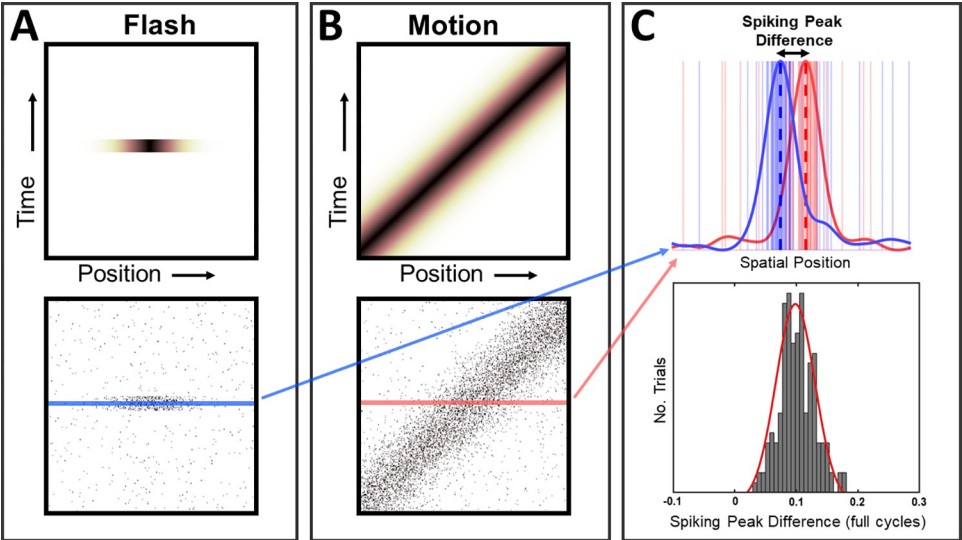

**Fig 2. Measuring spiking peaks in the static and velocity-tuned networks. A.** Example input layer activity in the static network, in response to a stationary flash. *Top*: Firing rate response of input layer neurons. *Bottom*: Spikes generated based on these underlying firing rates, which are transmitted via feedforward connections to layer 2, and so on. **B.** Example of input layer activity in a velocity-tuned subpopulation, in response to a motion stimulus. Top and bottom panels indicate firing rates and spikes, respectively, as in (A). **C.** Spiking distribution peak differences between static network responses to flash and velocity-tuned network responses to motion. *Top*: Spiking locations across neurons in a single layer (vertical lines) and timepoint are convolved with a Gaussian kernel to produce a continuous probability density function (PDF), in both the static network (blue) and the velocity-tuned network (red). A measure of spiking peak difference is calculated based on the angular error between peaks (dotted lines). *Bottom*: Spiking peak differences are recorded across 200 trials, and a von Mises distribution (red curve) is fitted to compute the average spiking peak difference.

stimulus at that timepoint, for a duration of 20 ms (Fig 2A). This stimulus is delivered as separate input to the static network, and a PDF is fitted to the resulting spiking activity to form an estimate of positional representation in the same manner as for motion responses in the velocity-tuned subpopulations (Fig 2C, blue lines).

## Data analysis

**Measuring receptive-field shift.**   The effect of STDP can be measured by the shift in neural receptive fields produced by the training procedure, comparing the mean receptive field of networks that have not undergone any training with the mean receptive field after training (for each of the velocity-tuned networks; Fig 3). To generate the mean receptive field of all neurons in a given layer, the discrete set of weights for each neuron is first converted into a continuous PDF via convolution with a wrapped Gaussian kernel (so that the magnitude of the weight for a given connection is related to the probability density at that point in space). PDFs for each neuron are then shifted to a common, arbitrary neural index, before being averaged to produce the mean tuning curve. This is performed for each velocity-tuned network, as well as for an equal number of untrained networks. Similar to the approach used in Burkitt et al [15]., the difference in the receptive-field center is measured for each trained-untrained network pair, resulting in measures of receptive-field shift for each of the velocity-tuned subpopulations.

**Finding peak of spiking response.**   In addition to analysis of receptive-field shifts, we also assess the distribution of spiking activity at each layer in order to probe stimulus location representations within the network. The population representation of stimulus position is

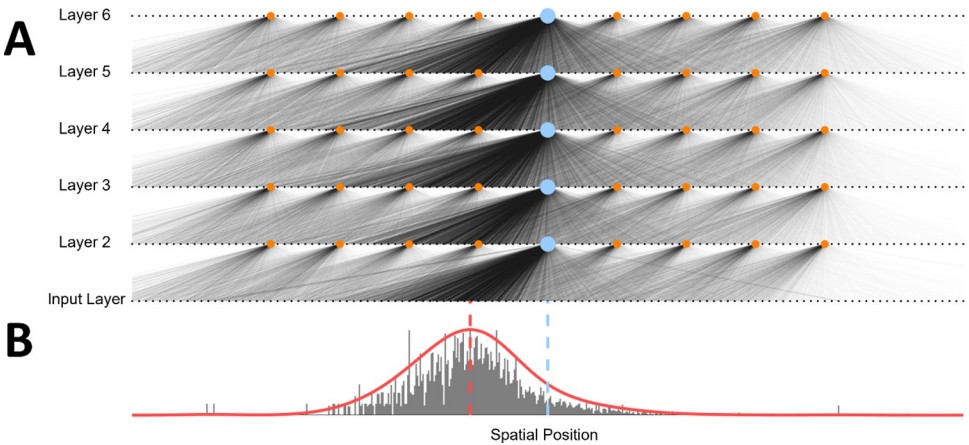

**Fig 3. Network connections in a trained network. A.** Subset of neural connections after STDP learning. Weights on the left-hand side of each receptive field (opposite to direction of motion) have been strengthened, while weights on the right-hand side are weakened overall. **B.** Distribution of feedforward weights for central neuron at layer two. The probability density function is now shifted in the opposite direction to motion, so that the peak of the tuning curve (dotted red line) is shifted away from the neuron's original receptive field center (blue).

estimated by computing the peak of the spiking distribution across neurons for a given layer and timepoint. To this end, a continuous PDF is generated by taking the sample of discrete spiking locations over the interval [0, 1] and convolving it with a wrapped Gaussian kernel to produce a continuous PDF across the range of neural positions. The neural index closest to the maximum of the PDF is taken as the peak of the spiking response for that layer and timepoint (Fig 2B).

**Measurement of network activity.** To measure the compensatory effect of STDP on moving object localisation, we compare spatial representations of stimulus position between the stationary and motion stimulus conditions at a given point in time. It is first necessary to decide at which timepoint the comparison between responses will be made, leaving enough time for the network to build up a response to the input. To determine the precise time-to-peak response for each layer, we use a simplified version of the network that propagates firing rates alone (without conversion to spiking activity) and recorded the temporal peak of activity for each layer of an untrained network, in response to a stationary stimulus with 20 ms duration.

Fig 4 shows the firing rates at each layer for neurons tuned to the flash position. The temporal peak at each layer is subject to two sources of delay: the *neural transmission delay* $t_{\text{delay}}$ which determines the length of time required for activity in one layer to reach the next, and an additional *activation delay* $a_{\text{delay}}$ which is due to the temporal integration required to build up a response, and depends primarily upon the parameter values of the passive membrane time constant and stimulus intensity. This method is used to derive the layer-specific timepoints which are used for comparison of spiking responses in the static network and each velocity-tuned network. For each trial of the simulation, the angular difference between the static and motion stimulus responses are recorded, and a von Mises distribution is fitted to the resulting vector of angular error across trials in order to estimate the mean angular error (Fig 2C, bottom panel). These analyses allow visualisation of the effect of STDP in two ways: in terms of the receptive field shift, via comparison of the trained velocity-subpopulations with their untrained counterparts, and in terms of positional representation shifts, via comparison of spiking responses across the velocity-tuned subpopulations and the static network.

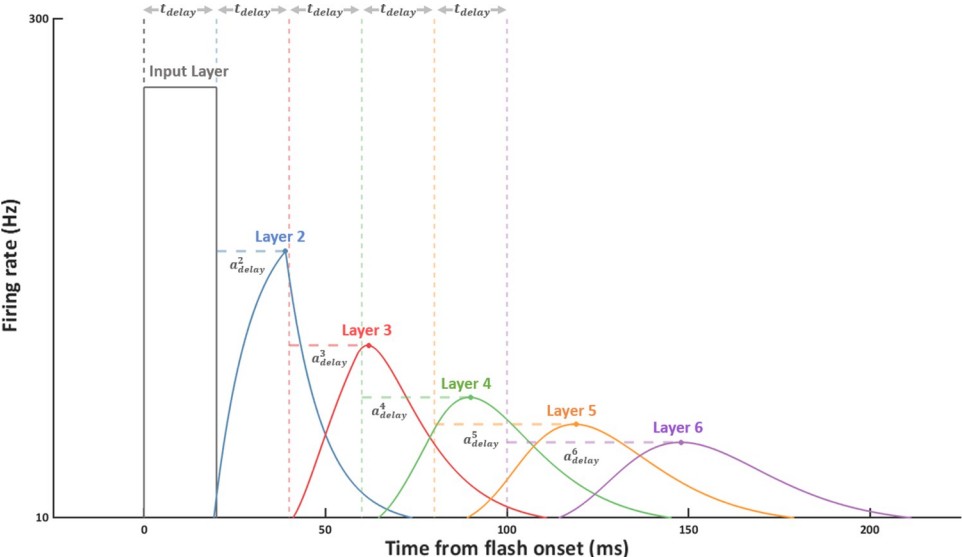

**Fig 4. Firing rate responses to 20 ms flash presentation.** The earliest point at which stimulus evoked activity can reach each higher layer is indicated by the dashed vertical lines (corresponding to a neural transmission delay $t_{\text{delay}}$ of 20 ms per layer). Additional time is required to reach a peak of activity beyond this point (dashed horizontal lines), denoted by $a_{\text{delay}}^{\text{l}}$, where l refers to the layer number. This additional activation delay is due to the temporal integration required to build up a response.

## Results

Fig 5A shows the degree of difference between receptive-field peaks measured before and after STDP training, across each value of membrane time constant. At the lowest velocity (0.1 cycles/s), receptive field shifts for higher layers beyond layer 2 were minimal. As velocity increased, the degree of shift increased for each layer, with a substantial difference in magnitude apparent between layer 2 and the subsequent layers. The magnitude of separation between layer 2 and the rest of the higher layers gradually reduced until around 0.8 cycle/s, at which point there was little variability in the degree of separation between each layer, although the general ordering of receptive field shift magnitude was maintained. While the first higher layer (layer 2) drove most of the overall receptive field shift at lower velocities, it leveled out around 0.6 cycles/s, while the shift at higher layers continued to increase beyond this velocity. Therefore, as velocity increased, the additional hierarchical levels had increasing contributions to the overall capacity of the network to push forward the represented location of the moving stimulus.

At the highest velocities (2 to 5 cycles/s), the trend reversed: increased velocity yielded progressively lower receptive field shifts. This was likely caused by the temporal integration of activity required to build up a response–due to the speed of the moving stimulus, there was less sustained activity within each neuron's receptive field that was required to drive STDP. In support of this conclusion, Fig 5B shows that as the membrane time constant increased (meaning more sustained activity was required to build up a response), a greater reduction in receptive field shift at the high velocities was observed. For velocities up to 1 cycle/s, the pattern of shifts across membrane time constants was similar, with a slight decrease in overall magnitude as layer number increased, particularly for the 5ms condition at lower velocities. At the high velocities, higher time constants led to lower receptive field shifts, an effect which became greater as velocity increased (Fig 5B, velocities 2 to 5). Inspection of the change in receptive-field center across training timepoints (Fig 6) indicates that the receptive-field center measured

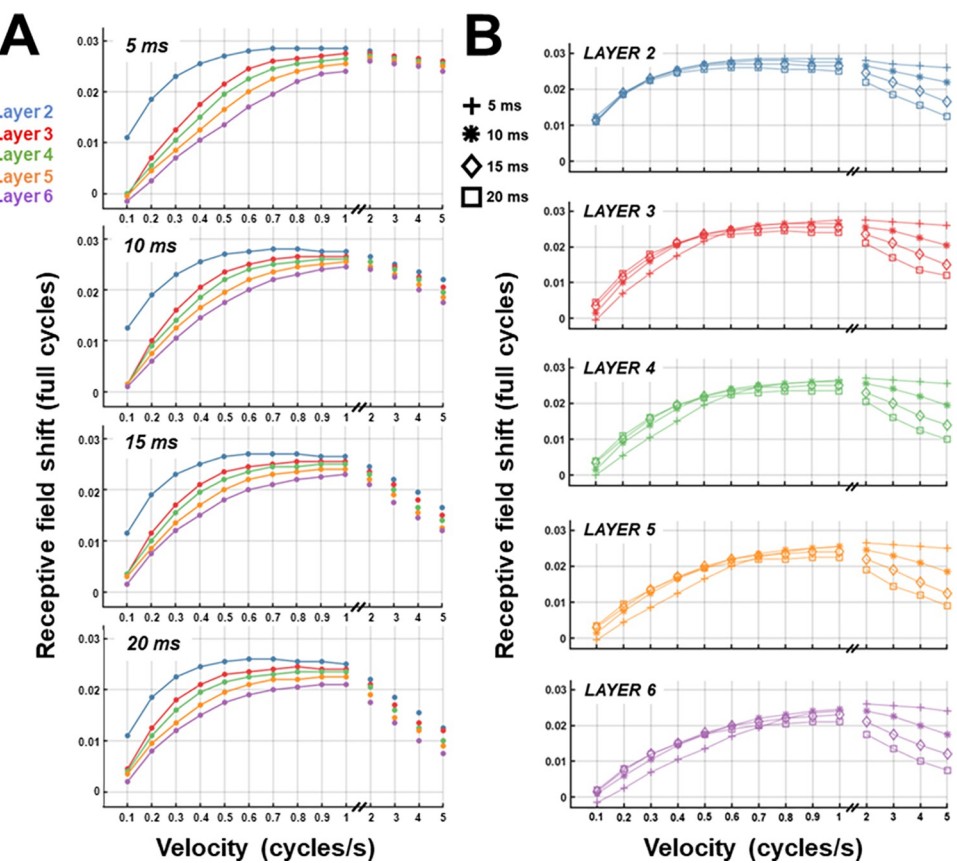

**Fig 5. Receptive field shifts caused by STDP training.** Magnitude of receptive field shift due to STDP learning, for each layer and velocity. Positive values indicate receptive field shift in the direction opposite to motion. **A.** Receptive field shifts by layer. Subplot titles indicate the value of the passive membrane time constant, with different layers indicated in different colours. **B.** Receptive field shifts by membrane time constant. Subplot titles indicate layers, and different time constants are differentiated by markers.

at the end of the training procedure reflected a stable, asymptotic level of shift produced by STDP.

A comparison of the simulated responses to static and moving stimuli was undertaken by directly inspecting the pattern of spiking behaviour across the static network and velocity-tuned networks. Fig 7 shows the spatial difference between population spiking distributions in response to the flash and motion conditions. As a consistency check, the magnitude of spiking peak differences for the input layer was approximately zero, indicating no difference between the flash and motion responses for this layer at the comparison timepoint (see *Measurement of network activity*). This indicates that the method of timepoint selection was successful in producing meaningful comparisons between the flash and motion responses. Positive peak differences were observed for layers 2 and above, which increased as a function of layer and velocity beyond 0.1 cycles/s. As with the pattern of receptive field shifts across velocity, the highest layers continued to produce substantial forward shifts in the represented location of the motion stimulus, even when the rate of increase at lower layers was minimal (consider the difference between magnitudes at the highest and lowest velocities for layer 6 compared to layer 1). This is presumably due to the fact that each layer inherits the receptive-field shifts from the preceding layer below it—the pattern of spiking distributions in response to motion at a given layer

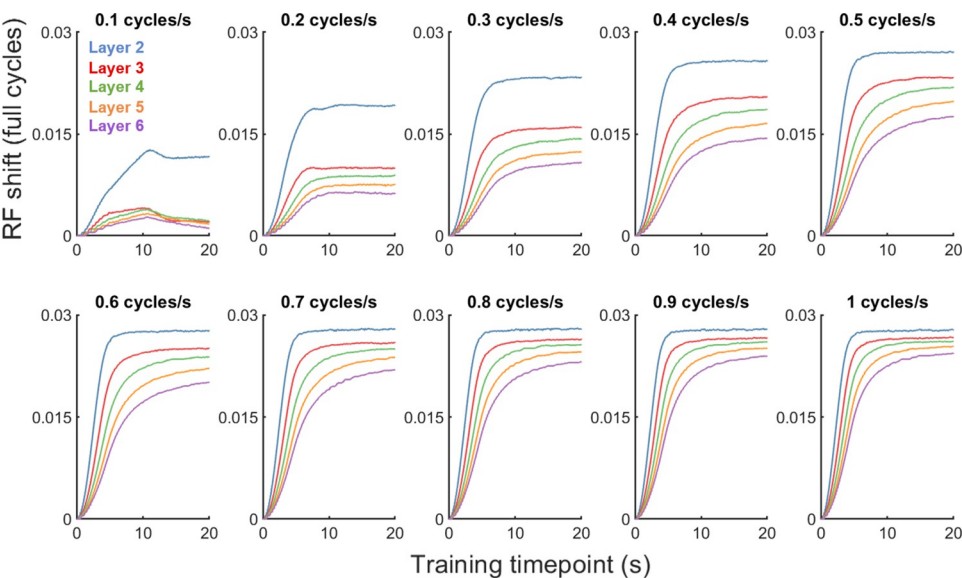

**Fig 6. Receptive field shift magnitudes across training timepoints.** The time-course of receptive field shifts during training, as a result of STDP.

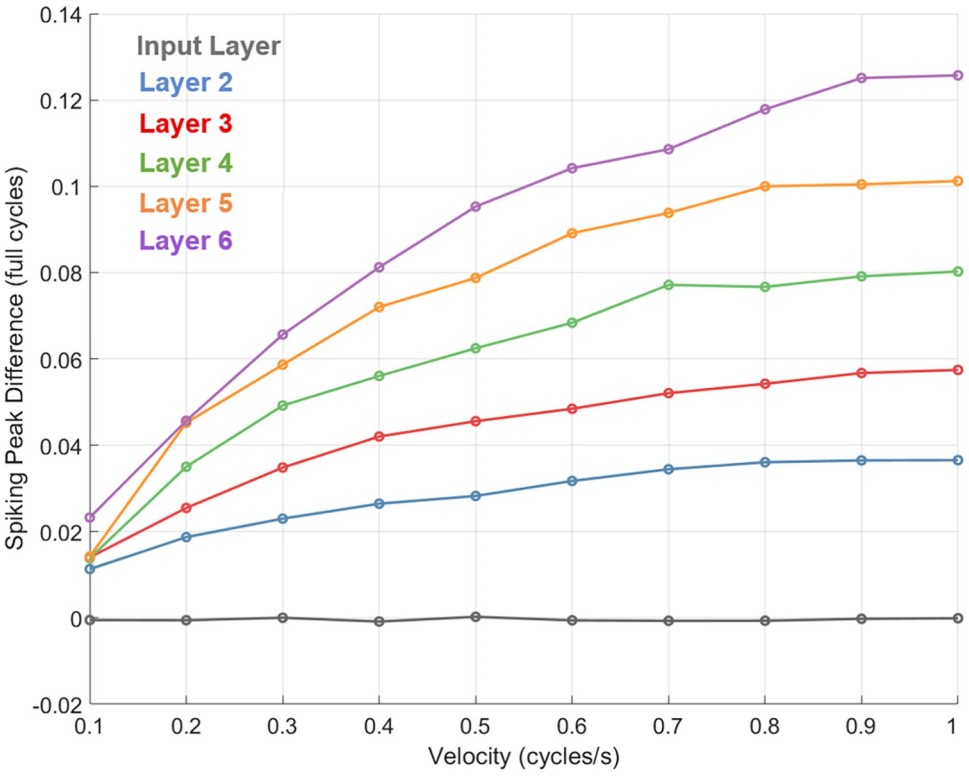

**Fig 7. Spiking peak differences between static and velocity-tuned networks.** Difference between peaks of spiking distributions in static network responses to a stationary stimulus versus velocity-tuned network responses to motion. Positive values indicate the represented location is shifted in the direction of motion. Results are shown for a 10 ms passive membrane time constant.

can therefore be seen as reflecting the accumulation of all receptive-field shifts up to and including that layer.

The results described above are based on comparison of the spiking responses to motion in the velocity-tuned networks, with the spiking response to a stationary flash in the static network–taking into account the time required for input to theoretically be accessible by each layer of the neural subpopulations. We were also interested in analysing the ongoing positional representations of each velocity-tuned network, relative to the realtime stimulus position–an analysis that more directly addresses the concept of 'representational lag'. To do so, we took the angular difference between the peak of the spiking distribution at each layer at the midpoint of each trial, relative to the stimulus location at that timepoint. Averaging these angular differences across 200 trials yielded a measure of the spatial difference between stimulus position and its represented location at the current point in time, for each layer and velocity.

Fig 8A plots these spatial differences for networks that have undergone STDP training, as well as for a set of untrained networks. In the untrained networks, increased velocity and layer depth both caused positional representations to increasingly lag behind the current stimulus position. In the velocity-tuned networks that underwent STDP training, the positional representations were shifted forward relative to the corresponding untrained networks. Representations at each of these layers were located ahead of the stimulus in the low velocities, and at or just behind the stimulus in the higher velocities. We also observed another effect: the degree of spatial separation across layers within each velocity-tuned network was reduced compared to the untrained networks. That is, the correspondence of positional representations across layers at a single point in time became greater in the trained networks. To quantify the magnitude of this 'temporal alignment' across layers, we took the average of the absolute spatial distance

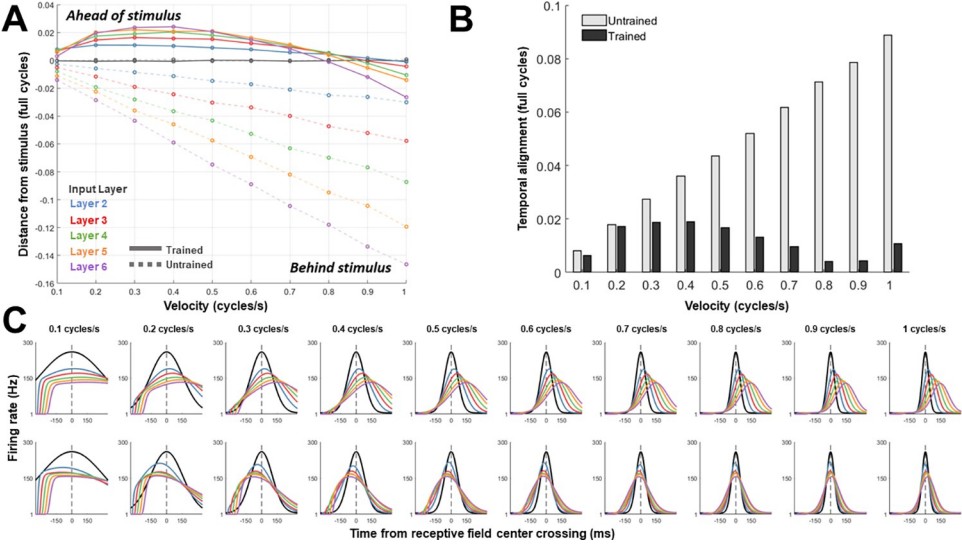

**Fig 8. Realtime position representations. A.** Distance between population representation of stimulus position at each layer, relative to its true position, at the midpoint of the motion trajectory. Positive values indicate a represented location that is ahead of the concurrent stimulus location. *Solid lines*: STDP-trained networks. *Dashed lines*: untrained networks. **B.** Alignment of layer-based representations, computed as the mean absolute deviation of all layers, across each velocity-tuned network and a corresponding set of untrained networks. Lower values indicate greater alignment of positional representations. **C.** Firing rates before and after STDP learning, for a single unit tuned to the central position of motion trajectories at each velocity tested. Due to the receptive-field shifts caused by STDP, the onset and/ or peak of the response at higher layers occurs before the input layer response, to varying degrees depending on velocity. These firing rates are determined via 700ms of motion input, in which only firing rates are transferred between layers (rather than spikes).

between representations at each of the higher layers (layers 2 and above) with the input layer, across each velocity. This measure produces values that are closer to zero when representations across layers are more aligned, and further from zero when representations are spread further apart.

Fig 8B shows this value for the trained and untrained networks. In the untrained networks, the degree of temporal alignment decreased predictably as a result of increased velocity. The networks that underwent STDP training showed a greater degree of alignment, which varied non-linearly as a function of velocity, and reached a minimum at 0.8 cycles/s. The alignment of activity across layers is also visible in the temporal responses of individual neurons to motion stimuli. Fig 8C shows the average firing rate during a single trial for the neuron at each layer tuned to the midpoint of the motion trajectory (to reduce simulation time, these figures were generated using firing rates alone, as modelling the precise timing of individual spikes is no longer required after STDP training has occurred). For the untrained networks, the peak response at each layer was delayed by the same amount across velocities. The width of the activity profile was narrower for higher velocities, reflecting the shortened time that the stimulus spent in each neuron's receptive field. In the case of the trained networks, responses at each layer were aligned much more closely in time, often peaking at or near the timepoint that the stimulus crossed the input layer neuron's receptive field center (Fig 8C, lower panels).

## Discussion

This study investigates the degree to which STDP learning effects observed in a single layer of simulated synaptic connections [15] are accumulated across six layers and using biologically realistic synapses. Furthermore, we explored the degree to which STDP-induced receptive field shifts, as a proposed mechanism for motion extrapolation, are able to reduce the spatial difference between the realtime position of a motion stimulus and its internal representation encoded across multiple levels of a hierarchical network with transmission delays. The time course of the postsynaptic response was incorporated into the model, in order to assess the capacity of the network to compensate for additional activation delays resulting from the temporal integration of synaptic input.

First, we show that the overall capacity of STDP to compensate for neural delays among feedforward connections of a visual network increases with the number of processing stages. Namely, the depth of the network increases the overall forward shift in the represented location of a moving stimulus; where receptive field shifts at lower layers reach an upper limit, shifts at higher layers continue to accumulate. However, contrary to the previous expectation that the same STDP-induced receptive field shifts occurring at the first set of synapses should take place in each subsequent level [15], we observed that the receptive field shifts of each level beyond layer 2 were decreased in magnitude, with the degree of difference decreasing as a function of velocity. Additionally, at the lowest velocity tested (0.1 cycles/s), receptive field shifts beyond layer 2 were negligible (roughly an order of magnitude smaller). The finding that STDP effects do not linearly accumulate with each additional layer is likely due to the widening of effective receptive fields at higher layers. Because neurons located at the outer edge of a higher layer neuron's receptive field receive input themselves from a spatially extended set of connections, the motion-evoked input to a neuron has a temporal window that widens progressively for each higher layer. Due to the STDP function, which produces smaller changes in weight for greater temporal delays between pre- and post-synaptic spikes, a wider temporal window of input will produce less overall receptive field shift. This would explain why as velocity increases, the degree of difference between two successive layers in the network decreases (in terms of receptive field shift), but never reaches or exceeds the magnitude of layer 2 (Fig 5).

The neural model used in the original study [15] was extended here by replacing the simplified instantaneous synaptic transmission mechanism with a more biologically realistic integration of synaptic input. As well as making the model more biologically realistic, this addition of temporal integration in the postsynaptic response adds another source of delay to the stimulus-evoked response of higher layer neurons. Although the peak magnitude of receptive field shift achieved in the current network is similar to that of the original model, it is reached at a lower velocity here (~1 cycle/s in the current model compared to 5 cycles/s in the original), indicating that the addition of temporal integration alters the temporal dynamics of the STDP weight evolution. In line with Burkitt et al [15]., we observed that the magnitude of receptive field shifts caused by STDP generally increase as a function of velocity. However, this effect is limited to a lower set of velocities: after approximately 1 cycle/s, receptive field shifts begin to decrease in magnitude. This attenuation of the effect is likely due to the fact that at higher velocities, the stimulus spends less time in each receptive field, meaning there is less sustained activity in a single location to build up activity and drive STDP. This interpretation is supported by the observation that longer membrane time constants cause this effect to become more pronounced (Fig 5B). Because longer time constants mean that neurons require a greater level of sustained activity to fire, the lack of sustained input will reduce the degree of STDP learning at these high velocities.

We observed that the magnitude of the forward shift in the encoded position of the motion stimulus was in some cases sufficient to lead to realtime representations of location that were ahead of or in line with the stimulus (Fig 8A), despite accumulated transmission delays in the network being as high as 100ms for layer six. It must be noted that this effect is dependent on the specific transmission delay and input velocity (as well as the membrane time constant and stimulus intensity). This is because the STDP mechanism modelled is insensitive to the specific transmission delay or input velocity. The observed increase in temporal alignment of positional representations across layers (Fig 8B) has the same dependency on delay time and velocity. In order to achieve temporal alignment across a broader range of velocities and transmission delays, it may be necessary to incorporate a feedback mechanism into the model, to allow extrapolated representations at higher levels to be adjusted based on their correspondence with lower-level activity. This notion was proposed in Hogendoorn et al [21]., who hypothesised that 'realtime temporal alignment' (RTTA) should occur in hierarchical networks with both feedforward and feedback connections. In their model, which extended the classical predictive coding model [22] to account for the presence of transmission delay times between hierarchical levels, prediction errors for motion stimuli are minimized when extrapolation mechanisms are implemented along both the feedforward and feedback pathways. An interesting property of this arrangement is that at any given point in time, neural populations across each hierarchical level represent the location of the moving stimulus in the same position, despite the transmission delay between levels. Note that for static inputs, such as a stationary object, extrapolation mechanisms would be not be required for RTTA to occur, given that the location of the represented stimulus is unchanging across time [21]. In the case of the static network presented here, representations at each layer would align with the position of a sustained, stationary stimulus (assuming enough time for each layer to receive input layer activity), given that no STDP training was performed on the static network weights (replicating the case in which STDP effects are spatially symmetric).

Although we have observed some degree of RTTA in this purely feedforward network, as mentioned the effect is limited in range. In order to provide support for the notion that RTTA is a fundamental property of the visual system, it should arise across a wide range of velocities and delay parameters. Further work could test this hypothesis by extending the model to incorporate a mechanism by which the degree of receptive field shift could be calibrated to

specific transmission delays and stimulus velocities. As suggested, the addition of feedback connections could allow extrapolated representations at higher levels to be adjusted based on their correspondence with subsequent input at lower levels, more akin to the type of predictive coding network described in Hogendoorn et al [21]. Additionally, incorporating multiple delay lines between neurons could potentially increase the capacity of RTTA to emerge in the network. Previous research has demonstrated how STDP may underlie 'delay selection': the selective potentiation of specific connections between neurons from a range of connections with varying delays [23,24], thought to occur early during development [25] and underlie processes requiring a high degree of temporal coherence between neurons [23]. Although we have modelled a single, fixed transmission delay between neurons, it would be possible to extend the model to incorporate a subset of connections between each pair of neurons, across which transmission delay time is varied. This would increase the range of adaptation in the temporal dynamics of the network [26], potentially expanding the range of velocities across which RTTA is observed. However, it is possible that such a network would still require some form of feedback in order to select delays that yield the highest degree of coherence across regions.

Previous work incorporating delay selection has addressed its capacity to generate motion selectivity in SNNs [27,28]. Related work has also highlighted the role of STDP as a mechanism by which motion-selectivity may arise [18,19,29]. In this work, we have not modelled the formation of motion encoding, but for simplicity have taken as a starting point discrete populations of neurons endowed with velocity-selectivity. An interesting avenue for future exploration would therefore be to also incorporate the formation of velocity-selectivity into the learning process, potentially via STDP-driven selection of specific feedforward delays.

In this work, we have included a simplified model of velocity-selectivity in which neurons are active in response to a singular, one-dimensional input velocity. This approach was adopted in line with the specific goal of addressing the effect of stimulus velocity on STDP-driven receptive field shifts, as a proof-of-principle, rather than deriving a model that captures the full extent of visual motion processing. Indeed, the model described here is likely to be largely inefficient for processing the full range of directions and speeds to which the visual system is sensitive. In addition, the one-dimensional nature of the network prevents its application to standard motion datasets, which are typically two-dimensional and also contain motion energy at various velocities. A more biologically realistic extension of this network would be to determine neural activity based on a characteristic tuning curve over a range of velocities. Future work could also extend the network to two-dimensions, and implement a global read-out mechanism that forms representations of a dynamic visual scene based on the weighted, simultaneous activity of all velocity-tuned networks.

The results of this study provide convergent evidence that STDP can cause shifting receptive fields in motion sensitive neurons [11,15], supporting its role as one of the potential mechanisms underlying forward-shifts in the represented location of moving objects. Whereas many studies are concerned with the role of local, horizontal circuitry underlying such shifts [9,10,12], the mechanism described here operates purely along feedforward pathways between retinotopic regions. While horizontal mechanisms of motion extrapolation described previously [30] could also be incorporated into the scheme presented here, we have chosen to focus purely on the effect of STDP on the feedforward pathway. Previous work has explored the role of asymmetries among feedforward connections between visual areas as a mechanism for motion for extrapolation [31,32], but typically leave open the question of how such asymmetries may form. For example, Kaplan et al. [31] describe a two-layer network that is able to compensate for a neural delay via anisotropic connectivity between neurons in each layer. In their model, delay compensation is achieved because activity at one layer is selectively transmitted to neurons in the second layer that will represent the correct location of stimulus, given

the velocity and delay time. The current study extends on this by addressing the question of how such an asymmetric connectivity pattern may emerge via STDP. Furthermore, this study also represents a first step toward understanding the cumulative effect of asymmetric feedforward connectivity across several visual areas.

In order to quantify the effect of STDP on instantaneous position representations of flashed and moving stimuli, we utilised an approach similar to a typical flash-lag effect (FLE) paradigm. In the FLE illusion, observers report a briefly flashed stimulus to lag behind a moving stimulus, despite their physical alignment [5]. A number of hypotheses regarding the underlying causes of the FLE have been proposed (for reviews see [33,34]), and our aim is not to advocate for any particular explanation of the FLE. The network described here, comprising only of feedforward connections and a simple learning mechanism, is not intended to capture the rich variety of mechanisms underlying perceptual illusions such as the FLE. For example, while we have shown how location representations can be shifted progressively forward as they ascend the visual hierarchy, it is probable that subsequent processes employing feedback pathways would further refine predictive signals engaged during tasks such as motion tracking. Predictive processing models [22,35,36] emphasise the role of higher level, model-based predictions which would extend the temporal and spatial range of motion extrapolation possible in the type of feedforward network described here. Our model, therefore, can be regarded as describing an initial, bottom-up predictive signal that occurs at the onset of retinal motion and plays a role in the very early stages of motion tracking. A rapid mechanism to shift forward the represented location of a moving object during the first feedforward sweep of activity would be advantageous to higher level predictive processes operating over greater temporal intervals. For example, tracking a fast-moving object that has entered the visual periphery, such as a clay shooter's target, would involve an early period of retinal motion prior to subsequent fixation and pursuit. The mechanism described here, occurring rapidly at the onset of retinal motion, would facilitate a quick saccade to the future location of the target. The over-extrapolation of stimulus location observed at lower velocities (Fig 8A) may indeed be computationally advantageous in this case, given the extra delays involved in initiation of the motor response.

In conclusion, we have shown that receptive field shifts caused by STDP accumulate in a multilayered network to produce forward shifts in the represented location of a moving stimulus well beyond that of a two-layer network. We have also shown that realtime positional representations across each layer of such a network become more aligned as a result of STDP learning. This mechanism involving purely feedforward pathways, constitutes a rapid predictive signal occurring at the onset of visual motion, and likely makes an important contribution to later, higher-level predictive processes.

## Author Contributions

**Conceptualization:** Charlie M. Sexton, Anthony N. Burkitt, Hinze Hogendoorn.

**Investigation:** Charlie M. Sexton.

**Methodology:** Charlie M. Sexton, Anthony N. Burkitt, Hinze Hogendoorn.

**Software:** Charlie M. Sexton, Anthony N. Burkitt, Hinze Hogendoorn.

**Supervision:** Anthony N. Burkitt, Hinze Hogendoorn.

**Visualization:** Charlie M. Sexton.

**Writing – original draft:** Charlie M. Sexton.

**Writing – review & editing:** Charlie M. Sexton, Anthony N. Burkitt, Hinze Hogendoorn.

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
