## [Decision Letter · Decision Letter 0]

27 Apr 2023

Dear Sexton,

Thank you very much for submitting your manuscript "Spike-timing dependent plasticity compensates for neural delays in a multi-layered network of motion-sensitive neurons" for consideration at PLOS Computational Biology.

As with all papers reviewed by the journal, your manuscript was reviewed by members of the editorial board and by several independent reviewers. In light of the reviews (below this email), we would like to invite the resubmission of a significantly-revised version that takes into account the reviewers' comments.

Reviewers appreciated the presentation of the scientific problem but raised several questions which require a major revision. Please update your manuscript and provide a response to the reviewers in order to consider a novel submission.

We cannot make any decision about publication until we have seen the revised manuscript and your response to the reviewers' comments. Your revised manuscript is also likely to be sent to reviewers for further evaluation.

Sincerely,

Laurent Udo Perrinet

Guest Editor

PLOS Computational Biology

Thomas Serre

Section Editor

PLOS Computational Biology

Reviewers appreciated the presentation of the scientific problem but raised several questions which require a ajor revision. Please update your manuscript and provide a response to the reviewers in order to consider a novel submission.

Reviewer's Responses to Questions

**Comments to the Authors:**

Reviewer #1: Overall I found the paper very interesting and quite well conceived. My remarks are mostly about minor details. This work uses a simple neuronal model (no homeostasis, inhibition...) but demonstrates the ability of STDP learning for compensating for delays during motion perception. Their related work mentions no similar work in that domain. The result part is thorough, but could benefit from some more quantitative analysis as it can be difficult to gather all the information from the plots only. All the results are presented on simulated inputs. It would be a great addition to try the model on real inputs or a standard motion dataset.

My major concerns are on the related work part. It presents very few papers on the subject except biological papers that focus on the biological effect of motion-lag compensation.

There is some literature missing on the subject of learning motion representations with spiking neural networks, such as:

- Unsupervised Learning of a Hierarchical Spiking Neural Network for Optical Flow Estimation: From Events to Global Motion Perception

Federico Paredes-Vallés, Kirk Y. W. Scheper, Guido C. H. E. de Croon

- Learning heterogeneous delays in a layer of spiking neurons for fast motion detection

Antoine Grimaldi and Laurent U Perrinet

- Spike timing-based unsupervised learn-ing of orientation, disparity, and motion representations in a spiking neural network.

T. Barbier, C. Teulière, and J. Triesch

Even though those papers do not focus specifically on the work of motion lag compensation, they present important mechanisms for motion tuning. If this is the only model that specifically implements a motion lag compensation framework, I would mention this explicitly.

I also have a few specific remarks as well as general questions that are of lesser concern:

line 139: Delays in biological model are not necessarily constant for each neuron. What problems, if any, do you expect for a model with variable delays ? What could be done to address such problems?

Line 210: The training occurs for around 20 seconds. This seems like a very short time compared to real biological training time. Please comment.

Fig 5: It is difficult to see the difference in shift between the 4 plots. Please provide a quantitative analysis.

You address the motion tuning as a dual layer phenomenon (lower cells encode position, higher cells encode motion). Have you thought about lower cells with synapses of various delays in order to encode motion instead? This could be worth discussing.

Reviewer #2: Following Fu et al (2004) observations, and Burkitt et al model (2021), the authors propose a model of receptive field shift in a feed forward neural network, under spike timing dependent plasticity (STDP). The model is composed of six layers of simple neuronal integrators with stochastic firing, and fixed axonal delays betwen layers. The neurons are organized on a ring, with a spatially-dependent connectivity. The first layer is stimulated with a moving stimulus at constant speed. The weights evolve under STDP with some compensatory (normalizing) mechanisms. The authors show a consistant shift of the receptive field of the neurons in the direction opposite to the motion, that accumulates over the layers. This non-supervised mechanism is shown to partly compensate the axonal delays accumulated over the layers.

The plasticity mechanism proposed only relies on a simple weight normalization, and remains blind to the true speed of the stimulus (overestimation for low velocities, underestimation for high velocities, with an "optimum" at around 0.8 cycle/s in fig. 9). As explained in the discussion, this mechanism alone is not enough to implement a true temporal alignment.

More problematically, the way the authors pretend their model may reflect the flash-lag effect (FLE) illusion is an overstatement, and the figure 2 is clearly misleading to that prospect. One can not pretend to propose a mechanistic model of the flash lag effect by using different weights for different stimuli (!).

In conclusion, the authors have provided a timely analysis of an important receptive field shift effect that is shown to cumulate over layers. However, the authors tend to oversell their results and falsely pretend that it compensates for the time delays (this is only true in a tiny interval), and may explain more high-level features like the flash lag effect.

For that reason, the paper can not be accepted in the present writing. Both the introducion and the descussion need to be rewritten to reflect the limited scope of the results.Figure 2 is clearly misleading and should be removed. Also fig. 8 should be removed for it suggest to a distracted reader that the temporal alignment is effective at all speed. Moreover, a new figure should clearly quantify, on a layer by layer basis, the shift in temporal alignment in function of the stimulus speed (for fig.9 is quite difficult to read from that perspective)

**Have the authors made all data and (if applicable) computational code underlying the findings in their manuscript fully available?**

Reviewer #1: Yes

Reviewer #2: Yes

PLOS authors have the option to publish the peer review history of their article (what does this mean?). If published, this will include your full peer review and any attached files.

Reviewer #1: No

Reviewer #2: **Yes: **Emmanuel Daucé
---

## [Decision Letter · Decision Letter 1]

21 Jul 2023

Dear Sexton,

Thank you very much for submitting your manuscript "Spike-timing dependent plasticity partially compensates for neural delays in a multi-layered network of motion-sensitive neurons" for consideration at PLOS Computational Biology. As with all papers reviewed by the journal, your manuscript was reviewed by members of the editorial board and by several independent reviewers. The reviewers appreciated the attention to an important topic. Based on the reviews, we are likely to accept this manuscript for publication, providing that you modify the manuscript according to the review recommendations.

Thanks for the revision of your paper. Feedback from reviewers suggest minor corrections to account for further comments.

Sincerely,

Laurent Udo Perrinet

Guest Editor

PLOS Computational Biology

Thomas Serre

Section Editor

PLOS Computational Biology

Thanks for the revision of your paper. Feedback from reviewers suggest minor corrections to account for further comments.

Reviewer's Responses to Questions

**Comments to the Authors:**

Reviewer #1: We thank the authors for their careful revision of the manuscript. All our concerns have been addressed satisfactorily. We think the manuscript is now ready for publication.

Reviewer #2: (no attachment)

I appreciated the effort made by the authors in significantly reworking the introduction and conclusions of this paper, which has gained clarity compared to the previous version. While clarifying their model and results, it seems to me that in their conclusions, the authors do not sufficiently discuss certain important limitations.

Firstly, the proposed model uses a large number of synaptic relays and layers of neurons for the processing of visual data. As stated, each speed selectively activates a sub-network and inhibits the rest. How can such a system claim to process a large number of directions and speeds (beyond those analyzed in the paper)? It appears highly improbable that the visual system dedicates a specialized system to each direction/speed, considering that the number of considered layers (six synaptic contacts) can account for the entirety of rapid visual processing (Thorpe et al., 1996). Would it be possible to do away with these specialized sub-networks to achieve more efficient and economical visual processing?

Secondly, the case of static inputs is superficially addressed. With the proposed model, temporal alignment is fine for targets with a speed of 0.8 cycles/s, but in principle, it should also be fine for stationary targets. Can the authors confirm whether a static target would be correctly represented in all layers, just like a moving target? Does the widening of the receptive field with layers for low speeds pose a problem? If so, how can it be overcome? How can a system be proposed that accurately processes both static and moving targets across layers?

Please elaborate on your conclusions and discussion regarding these two points.

Minor point: Figure 2 remains unclear and should be improved, perhaps by reversing the black/white contrast. In the figure caption, it is difficult to understand how many layers were used to measure the temporal shift (layer 2 and above??). Some axes lack units.

**Have the authors made all data and (if applicable) computational code underlying the findings in their manuscript fully available?**

Reviewer #1: Yes

Reviewer #2: **No: **I may be wrong, but it seems that the code of the model was not made available.

PLOS authors have the option to publish the peer review history of their article (what does this mean?). If published, this will include your full peer review and any attached files.

Reviewer #1: No

Reviewer #2: No

Figure Files:

Data Requirements:

Reproducibility:

References:

---

## [Editor Report · Decision Letter 2]

22 Aug 2023

Dear Sexton,

We are pleased to inform you that your manuscript 'Spike-timing dependent plasticity partially compensates for neural delays in a multi-layered network of motion-sensitive neurons' has been provisionally accepted for publication in PLOS Computational Biology.

We would like to thank you for your updated manuscript which correctly answered the comments of reviewers. I appreciated the availability of the code despite the fact that I could not test it myself. Note that the reference to the conference paper "Grimaldi A, Perrinet LU. Learning hetero-synaptic delays for motion detection in a single layer of spiking neurons. 2022." is now published as "Grimaldi A, Perrinet LU. Learning heterogeneous delays in a layer of spiking neurons for fast motion detection. 2023. Biological Cybernetics."

Best regards,

Laurent Udo Perrinet

Guest Editor

PLOS Computational Biology

Thomas Serre

Section Editor

PLOS Computational Biology

---

## [Editor Report · Acceptance letter]

1 Sep 2023

PCOMPBIOL-D-23-00284R2 

Spike-timing dependent plasticity partially compensates for neural delays in a multi-layered network of motion-sensitive neurons

Dear Dr Sexton,

I am pleased to inform you that your manuscript has been formally accepted for publication in PLOS Computational Biology. Your manuscript is now with our production department and you will be notified of the publication date in due course.

With kind regards,

Zsofi Zombor
